# VONet: Unsupervised Video Object Learning With Parallel U-Net Attention and Object-wise Sequential VAE

**Haonan Yu**
Horizon Robotics
Cupertino, CA 95014, USA
`haonan.yu@horizon.cc`

**Wei Xu**
Horizon Robotics
Cupertino, CA 95014, USA
`wei.xu@horizon.cc`

## Abstract

Unsupervised video object learning seeks to decompose video scenes into structural object representations without any supervision from depth, optical flow, or segmentation. We present VONet, an innovative approach that is inspired by MONet. While utilizing a U-Net architecture, VONet employs an efficient and effective parallel attention inference process, generating attention masks for all slots simultaneously. Additionally, to enhance the temporal consistency of each mask across consecutive video frames, VONet develops an object-wise sequential VAE framework. The integration of these innovative encoder-side techniques, in conjunction with an expressive transformer-based decoder, establishes VONet as the leading unsupervised method for object learning across five MOVI datasets, encompassing videos of diverse complexities. Code is available at https://github.com/hnyu/vonet.

## 1 Introduction

Unsupervised video object learning has garnered increasing attention in recent years. It focuses on the extraction of structural object representations from video sequences, without the aid of any supervision, such as depth information, optical flow, or labeled segmentation masks. The goal is to enable machines to automatically learn to discern and understand objects within video streams, an essential capability with wide-ranging applications in fields such as autonomous robotics (Veerapaneni et al., 2020; Creswell et al., 2021), surveillance (Jiang et al., 2019), and video content analysis (Zhou et al., 2022). The utilization of such object-centric representations could lead to improved sample efficiency, robustness, and generalization to novel tasks (Greff et al., 2020).

Slot attention methods (Locatello et al., 2020; Kipf et al., 2021; Elsayed et al., 2022; Singh et al., 2022b) have recently demonstrated significant successes in video object learning. They typically utilize a CNN to extract a feature map from an input image. This feature map is spatially flattened into a sequence of features, which are then queried by each slot latent to generate an attention mask. Our observation is that slot attention often encounters a dilemma, referred to as "granularity versus continuity". To generate fine-grained attention masks, it is necessary to select a large spatial shape for the feature map. However, doing so makes it challenging to ensure the smoothness of the mask due to the nature of the Key-Query-Value attention mechanism (Locatello et al., 2020). Sometimes the absence of smoothness may result in significant mask quality degradation.

This paper introduces VONet for unsupervised video object learning. Inspired by MONet (Burgess et al., 2019) for image object learning, we posit that the inductive bias for spatial locality, as seen in the U-Net (Ronneberger et al., 2015) of MONet, offers a solution to the dilemma. However, MONet's recurrent attention generation, forwarding the same U-Net multiple times sequentially, is very inefficient when handling a large number of slots, and consequently impedes its further application to video. Our first key innovation is an efficient and effective parallel attention inference process (Figure 1, b) that generates attention masks for all slots simultaneously from a U-Net. It can sustain a nearly constant inference time as the number of slots increases within a reasonable range.

Furthermore, to achieve temporal consistency of objects between adjacent video frames, VONet incorporates an object-wise sequential VAE framework. This framework adapts the conventional

sequential VAE (Kingma & Welling, 2013; Hafner et al., 2019) to the context of multi-object interaction and dynamics. The minimization of the KLD between the posterior and a forecasted prior models the dynamic interaction and coevolvement of multiple objects in the scene. This encourages the emergence of slot content that is temporally predictable and thus consistent in a holistic manner. By adjusting the weight of the KLD, we are able to control the importance of temporal consistency relative to video reconstruction quality.

To further bolster its capabilities, VONet employs an expressive transformer-based decoder (Singh et al., 2022b) that empowers itself to successfully derive object representations from complex video scenes. To showcase the effectiveness of VONet, we conduct extensive evaluations across five MOVI datasets (Greff et al., 2022) encompassing video scenes of varying complexities. The evaluation results position VONet as the new state-of-the-art unsupervised method for video object learning.

## 2 RELATED WORK

Numerous prior studies, such as Burgess et al. (2019); Greff et al. (2019); Locatello et al. (2020); Engelcke et al. (2021); Singh et al. (2022a); Zoran et al. (2021); Emami et al. (2021); Hénaff et al. (2022); Seitzer et al. (2022), explored unsupervised object learning in single images. For unsupervised video object learning, applying these image-based methods to video frames independently is not a viable approach, as it would likely result in slot masks lacking temporal consistency. A conventional strategy for transitioning from image object learning to video object learning entails inheriting and modifying the slot content from the preceding time step. For instance, AIR (Eslami et al., 2016), SQAIR (Kosiorek et al., 2018), STOVE (Kossen et al., 2019), and SCALOR (Jiang et al., 2019) all employed a propagation process in which a subset of currently existing objects is propagated to the next time step; TBA (He et al., 2019), ViMON (Weis et al., 2021), SAVI (Kipf et al., 2021), SAVI++ (Elsayed et al., 2022), STEVE (Singh et al., 2022b), VideoSAUR (Zadaianchuk et al., 2023), and RSM (Nguyen et al., 2024) initialized slots for the current step using the output of a forward predictor/tracker applied to the preceding slots. Another technique for ensuring temporal consistency is to model constant object latents across time, as demonstrated in Kabra et al. (2021). These object latents remain invariant across all frames by design, enabling stable object tracking. Alternatively, an explicit temporal consistency loss could be incorporated. Creswell et al. (2021) proposed an alignment loss which ensures that each object is represented in the same slot across time; Bao et al. (2022) encouraged similarity between the feature representations of slots in consecutive frames. Our approach inherits preceding slot content while also introducing a KLD loss of a sequential VAE to further enhance temporal consistency.

Due to the absence of supervision signals, most existing methods could only handle uncomplicated video scenes with uniformly-colored objects or objects with simple sizes and shapes. For instance, in an unsupervised setting, Creswell et al. (2021); Kipf et al. (2021); Elsayed et al. (2022) demonstrated effectiveness primarily on uniformly-colored objects with clean backgrounds. Similarly, while SCALOR (Jiang et al., 2019) showcases an impressive capability in discovering and tracking dozens of simple objects of similar sizes in certain videos, it exhibits sensitivity to hyperparameters related to object scale and size ratio. As a result, it performs inadequately when dealing with textured objects of diverse shapes and sizes. STEVE (Singh et al., 2022b), on the other hand, successfully improved performance in complex videos through the introduction of an expressive transformer-based decoder. Nevertheless, it faces significant issues of over-segmentation and object identity swapping in less complex videos. Our method adopts the same transformer-based decoder to handle complex video scenes, while still being able to maintain superior performance in simple scenes.

A substantial body of related work in the field of video segmentation (Zhou et al., 2022) relies on supervision signals such as segmentation masks, depth information, optical flow, and more. Our problem also bears relevance to the domain of video object tracking (Ciaparrone et al., 2020), where object locations or bounding boxes are specified in the initial video frames and tracked across subsequent frames. Since we do not assume these additional learning signals, due to space constraints, we will not delve into detailed discussions of these two topics here. Notably, certain previous works, such as Zadaianchuk et al. (2023), employ a vision backbone pretrained on large-scale datasets for feature extraction from video frames, aiding in the process of object learning. While their objective remains unsupervised, the use of pretrained features integrates valuable prior knowledge of everyday object appearances. Consequently, the results they reported, though surpassing what typical unsupervised methods including ours could achieve, hold limited reference value in this context.

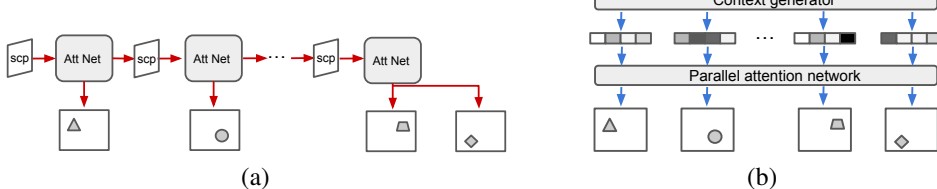

(a)                                                                      (b)

Figure 1: Attention processes of MONet (a) and VONet (b) on a single image. Red arrows represent sequential operations ("scp" stands for the MONet scope), while blue arrows at the same horizontal level represent parallel operations. The dependency on the input image has been omitted for clarity.

VONet shares a similarity with ViMON (Weis et al., 2021) in that both extend MONet (Burgess et al., 2019) from images to video. However, several major differences exist between the two: i) VONet parallelizes the original MONet architecture for efficient inference while ViMON still generates slots recurrently. ii) VONet seeds the attention network with context vectors while ViMON does this using previous attention masks. iii) VONet formulates an object-wise sequential VAE to promote temporal consistency, whereas ViMON ignores this by using a typical intra-step VAE.

## 3 PRELIMINARIES

**MONet for unsupervised image object learning.** The Multi-Object network (MONet) (Burgess et al., 2019) is a scene decomposition model designed for single images. A forward pass of MONet generally consists of two stages: mask generation and image reconstruction with a component VAE. Given an input RGB image $\mathbf{x} \in \mathbb{R}^{H \times W \times C}$, MONet utilizes a recurrent procedure to sequentially generate $K$ attention masks $\mathbf{m}_k \in [0, 1]^{H \times W}$, with $K$ representing the predefined number of slots. A *slot* represents either an object or a background region. The procedure comprises $K - 1$ steps, and at each step, the mask is determined as $\mathbf{m}_k = \mathbf{s}_{k-1}\alpha(\mathbf{x}, \mathbf{s}_{k-1})$, where the scope $\mathbf{s}_{k-1} \in [0, 1]^{H \times W}$ represents the remaining attention for each pixel, with the initial scope being $\mathbf{s}_0 = \mathbf{1}$. The attention network, denoted as $\alpha$, is implemented as a U-Net (Ronneberger et al., 2015), to predict the amount of attention the $k$-th step will consume from the current scope $\mathbf{s}_{k-1}$. The scope is then updated as $\mathbf{s}_k = \mathbf{s}_{k-1}(\mathbf{1} - \alpha(\mathbf{x}, \mathbf{s}_{k-1}))$. At the last step, MONet directly sets $\mathbf{m}_K = \mathbf{s}_{K-1}$, which ensures that the entire image is explained by all slots ($\sum_{k=1}^{K} \mathbf{m}_k = \mathbf{1}$). Conditioned on the predicted attention mask, each slot undergoes independent processing through a VAE to (partially) reconstruct the input image. The VAE first encodes each slot into a compact embedding by $\mathbf{z}_k \sim q(\mathbf{z}_k|\mathbf{x}, \mathbf{m}_k)$ and then derives a decoded image distribution $o(\mathbf{x}|\mathbf{z}_k)$ from this embedding. To do so, pixels $\{\mathbf{x}_n\}$ are decoded independently from each other conditioned on the slot embedding, namely $o(\mathbf{x}|\mathbf{z}_k) = \prod_n o(\mathbf{x}_n|\mathbf{z}_k)$. Finally, to consolidate the reconstruction results from different slots, MONet formulates a mixture of components decoder distribution in its training loss [1]:

$$\mathcal{L}_{\text{MONet}} = - \sum_{n=1}^{H \times W} \log \sum_{k=1}^{K} o(\mathbf{x}_n|\mathbf{z}_k)\mathbf{m}_{k,n} + \beta \sum_{k=1}^{K} D_{KL}\Big(q(\mathbf{z}_k|\mathbf{x}, \mathbf{m}_k)\Big|\Big|p(\mathbf{z}_k)\Big), \tag{1}$$

where the prior $p(\mathbf{z}_k)$ is a unit Gaussian. Intuitively, each slot only needs to encode/decode image regions that has been selected by its attention mask. Figure 1 (a) depicts the simplified attention process of MONet, emphasizing the data flow during a forward pass. MONet is deterministic and its masks have only unidirectional dependencies. Due to the recurrent nature of mask generation, as $K$ increases, the inference time for a forward pass will become prohibitive.

**Unsupervised video object learning.** In the context of unsupervised video object learning, each input sample is a sequence of RGB frames $\{\mathbf{x}_1, \ldots, \mathbf{x}_T\}$. The goal is to determine a set of attention masks, $\{\mathbf{m}_{t,k}\}_{k=1}^{K}$, for each frame $\mathbf{x}_t$ as in the single-image case. Additionally, the mask sequence, $\{\mathbf{m}_{t,k}\}_{t=1}^{T}$, associated with a specific slot $k$, should focus on a consistent set of objects. Because MONet was originally proposed for single images, there is no guarantee that directly applying it to two adjacent video frames will result in temporally coherent object masks, even though the two video frames might look similar. It is yet to be determined how MONet can be extended to facilitate video object learning with temporal consistency.

---

[1]MONet included a third loss term to encourage the geometrical simplicity of the attention masks $\mathbf{m}_k$ by having the decoder also reconstruct these masks. This loss is not discussed here.

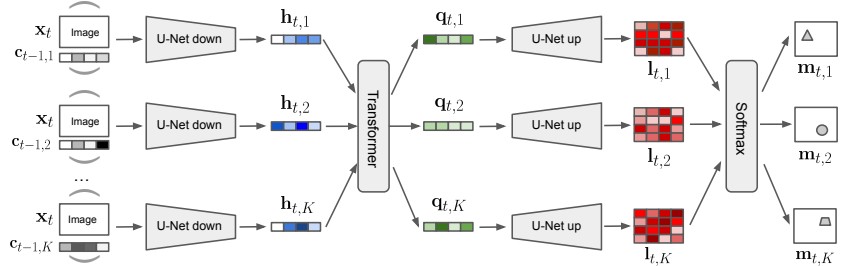

Figure 2: Diagram of the parallel attention network. Except for the transformer and softmax operator, it is possible to parallelize the executions related to the U-Net components. The skip connections between the U-Net downsampling and upsampling layers have been omitted for clarity.

## 4  VONET FOR UNSUPERVISED VIDEO OBJECT LEARNING

**Parallel U-Net attention.** VONet begins by eliminating the concept of "scope" in MONet and introduces *context*, a compact embedding vector expected to encompass prior object information for each slot to be generated. Let the context vector of slot $k$ at time step $t$ be $\mathbf{c}_{t-1,k}$. Instead of recurrent mask generation, VONet generates all masks at step $t$ at once (Figure 1, b):

$$\mathbf{m}_{t,1}, \ldots, \mathbf{m}_{t,K} = \text{ParallelAttn}(\mathbf{x}_t, \mathbf{c}_{t-1,1}, \ldots, \mathbf{c}_{t-1,K}). \tag{2}$$

The parallel attention network operates by simultaneously applying the same U-Net architecture to the $K$ context inputs in parallel, while establishing communication among the slots at the U-Net bottleneck layer. To achieve this, the output of the U-Net downsampling path is first flattened, and the outputs from different slots are pooled to create a sequence. This sequence is then input into a decoder-only transformer that produces a sequence of latent mask embeddings. Each mask embedding is further projected and reshaped to a 2D feature map, which then serves as the input for the U-Net upsampling path. In the final step, the $K$ output logits maps undergo pixel-wise softmax to derive the ultimate attention masks (Figure 2). Formally,

$$\begin{aligned} \mathbf{h}_{t,k} &= \text{U}_{\text{down}}(\mathbf{x}_t, \mathbf{c}_{t-1,k}), & \mathbf{q}_{t,1}, \ldots, \mathbf{q}_{t,K} &= \text{Transformer}_{\text{mask}}(\mathbf{h}_{t,1}, \ldots, \mathbf{h}_{t,K}). \\ \mathbf{l}_{t,k} &= \text{U}_{\text{up}}(\mathbf{q}_{t,k}), & \mathbf{m}_{t,1}, \ldots, \mathbf{m}_{t,K} &= \text{Softmax}(\mathbf{l}_{t,1}, \ldots, \mathbf{l}_{t,K}). \end{aligned} \tag{3}$$

Due to the single shared downsampling/upsampling path among the slots, in practice, $\text{U}_{\text{down}}$ and $\text{U}_{\text{up}}$ can be executed efficiently by rearranging the slot dimension into the batch dimension.

**Calculating the contexts.** One may pose the question: how can the context vectors be acquired? As previously mentioned, a context vector for a specific slot should encapsulate prior information regarding the aspects of the scene that this particular slot is intended to encode. Indeed, this prior information can naturally be derived from the content of that slot at the preceding time steps. For each slot $k$, we derive its context $\mathbf{c}_{t,k}$ (used by $t+1$) based on the history of the slot up to step $t$:

$$\mathbf{y}_{t,k} = \text{SlotEnc}(\mathbf{x}_t, \mathbf{m}_{t,k}), \quad \mathbf{r}_{t,k} = \text{RNN}(\mathbf{y}_{t,k}, \mathbf{r}_{t-1,k}), \quad \mathbf{c}_{t,k} = \text{MLP}_{\text{cxt}}(\mathbf{r}_{t,k}), \tag{4}$$

where the *per-frame* slot latent $\mathbf{y}_{t,k}$ is extracted through a *slot encoder* that takes both the image $\mathbf{x}_t$ and the generated attention mask $\mathbf{m}_{t,k}$ as the inputs. Meanwhile, $\mathbf{r}_{t,k}$ can be viewed as the *per-trajectory* slot latent, as it accumulates previous per-frame slot latents via a recurrent network. We initialize the per-trajectory slot latent by transforming a collection of noise vectors independently drawn from a unit Gaussian:

$$\mathbf{r}_{0,1}, \ldots, \mathbf{r}_{0,K} = \text{Transformer}_{\text{slot}}(\epsilon_1, \ldots, \epsilon_K), \quad \epsilon_k \sim \mathcal{N}(\mathbf{0}, \mathbf{1}), \tag{5}$$

This initialization signifies a stochastic process where the slots are collaboratively seeded prior to any exposure to the video content. Figure 9 (Appendix A.2) illustrates the temporal unrolling of VONet during a forward pass on a video.

**Object-wise sequential VAE.** Finally, we compute the slot posterior distribution directly based on its most recent per-trajectory slot latent at $t$ as

$$\mathbf{z}_{t,k} \sim q(\mathbf{z}_{t,k} | \mathbf{r}_{t,k}). \tag{6}$$

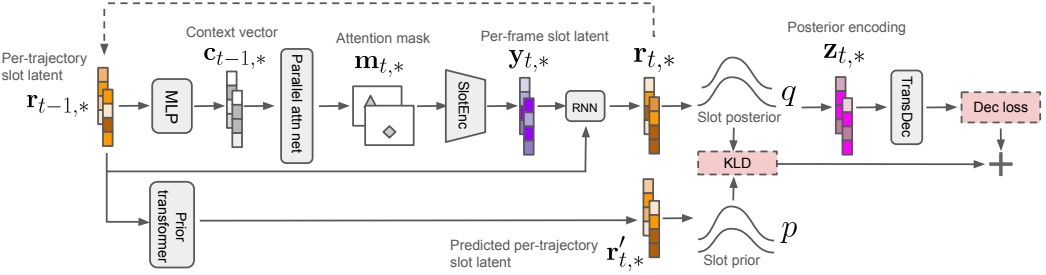

Figure 3: VONet's architecture. The dependency on the input image $\mathbf{x}_t$ has been omitted for clarity. * in the subscripts represents the collection of $K$ ($K = 2$ here) slots in parallel.

Meanwhile, we derive the slot prior distribution using all the $K$ per-trajectory slot latents up to $t-1$,

$$p(\bar{\mathbf{z}}_{t,k}|\mathbf{r}_{t-1,1}, \ldots, \mathbf{r}_{t-1,K}),$$

which in fact results in an object-wise sequential VAE. In essence, for proper learning of the prior, VONet must anticipate how each slot will evolve in interaction with the other $K - 1$ slots. A straightforward approach to instantiating the prior would involve utilizing a transformer to predict a future slot latent for each slot. This prediction can then serve as the basis for computing the prior:

$$\mathbf{r}'_{t,1}, \ldots, \mathbf{r}'_{t,K} = \text{Transformer}_{\text{prior}}(\mathbf{r}_{t-1,1}, \ldots, \mathbf{r}_{t-1,K}), \quad \bar{\mathbf{z}}_{t,k} \sim p(\bar{\mathbf{z}}_{t,k}|\mathbf{r}'_{t,k}). \tag{7}$$

For reconstruction, we opt for the transformer-based decoder (Singh et al., 2022a;b), owing to its remarkable performance observed in handling complex images. Thus our overall training loss is

$$\mathcal{L}_{\text{VONet}} = \sum_{t=1}^{T}\Big[ -\log P_{\text{TransDec}}(\mathbf{x}_t|\mathbf{z}_{t,1}, \ldots, \mathbf{z}_{t,K})$$
$$+ \beta \sum_{k=1}^{K} D_{KL}\Big(q(\mathbf{z}_{t,k}|\mathbf{r}_{t,k})\Big\|p(\mathbf{z}_{t,k}|\mathbf{r}_{t-1,1}, \ldots, \mathbf{r}_{t-1,K})\Big)\Big]. \tag{8}$$

Note that our KLD term, which integrates a learned prior, plays a pivotal role in strengthening the temporal consistency of individual slots across consecutive video frames. The rationale behind this enhancement lies in the fact that only slot representations that exhibit temporal consistency can exhibit predictability, consequently resulting in a reduced KLD loss.

An overview of the architecture of VONet is illustrated in Figure 3.

## 5 IMPLEMENTATION

**Backbone.** In the very initial stage of the encoding step, we use a CNN backbone (Appendix A.2) to extract a feature map from each input image $\mathbf{x}_t$ . The output from the backbone is shared between the parallel attention network (Eq. 3) and the slot encoder (Eq. 4), serving as the actual image input. Through parameter sharing, this backbone effectively reduces the total number of model parameters. It is trained from scratch using VONet's training loss.

**Parallel attention network.** Directly training the attention network formulated in Eq. 3 could be challenging due to the very depth of the U-Net. In each individual frame, this challenge stems from the intricate path that the gradient signal must traverse, starting from the VAE decoder, progressing through the slot encoder, then navigating the U-Net, and ultimately reaching the context vectors. Consequently, the generated masks $\mathbf{m}_{t,k}$ may be trapped in a trivial solution, where each pixel receives equal attention values from the $K$ slots. To address this issue, we first convolve each context vector across the backbone feature locations, yielding an estimated attention mask for each slot. This operation is analogous to slot attention (Locatello et al., 2020; Kipf et al., 2021). Subsequently, the U-Net is tasked with generating only delta changes (in log space) atop the estimated mask. This effectively establishes a special skip connection between the U-Net's input and output layers. We found that this skip connection plays a crucial role in learning meaningful attention masks.

**Slot encoder.** We adopt a simple architecture for the slot encoder. The attention mask is first element-wise multiplied with the backbone feature map, with broadcasting in the channel dimension.

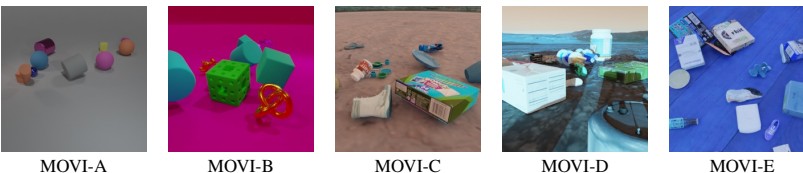

| MOVI-A | MOVI-B | MOVI-C | MOVI-D | MOVI-E |

Figure 4: Example video frames of the MOVI datasets. A,B,C contain up to 10 objects while D,E contain up to 23 objects in each video.

Then, the per-frame slot latent $\mathbf{y}_{t,k}$ is obtained by performing average pooling on the masked feature map across the spatial domain. While there may be other possible implementations of the slot encoder, we have found this straightforward approach to be highly effective and easy to train.

**Computing the per-trajectory slot latent.** We instantiate the RNN for calculating $\mathbf{r}_{t,k}$ (Eq. 4) as a GRU (Cho et al., 2014) followed by an MLP. Specifically, it is defined as

$$\mathbf{r}'_{t,k} = \text{GRU}(\mathbf{y}_{t,k}, \mathbf{r}_{t-1,k}), \quad \mathbf{r}_{t,k} = \text{LayerNorm}(\mathbf{r}'_{t,k} + \text{MLP}_{\text{slot}}(\mathbf{r}'_{t,k})). \tag{9}$$

LayerNorm (Ba et al., 2016) ensures that the latent values will not explode even for a long video.

**Computing the KLD.** We model both the posterior and prior distributions as diagonal Gaussians. When minimizing the KLD loss in Eq. 8, two primary influences come into play: the posterior is regularized by the prior, and conversely, the prior is shaped towards the posterior. These dynamics jointly contribute to the learning of slot representations. Following Hafner et al. (2020), we use a KL balancing coefficient $\kappa > 0.5$ to accelerate the learning of the prior relative to the posterior. This is done to prevent the regularization of the posterior by a poorly trained prior. Mathematically,

$$\text{KLD} = \kappa \cdot D_{KL}(\text{StopGrad}(q)\|p) + (1 - \kappa) \cdot D_{KL}(q\|\text{StopGrad}(p)). \tag{10}$$

## 6 EXPERIMENTS

**Benchmarks.** We assess VONet on five well-established public datasets MOVI-{A,B,C,D,E} (Greff et al., 2022), which include both synthetic and naturalistic videos. MOVI-A and MOVI-B consist of simple scenarios featuring nearly uniform backgrounds and objects with varying colors but minimal texture. MOVI-B exhibits greater complexity compared to MOVI-A, owing to the inclusion of 8 additional object shapes. MOVI-{C,D,E} stand out as the most challenging, featuring realistic, intricately textured everyday objects and backgrounds. While MOVI-{A,B,C} include up to 10 objects in a given scene, MOVI-D and MOVI-E include up to 23 objects. Each video within the five datasets comprises 24 frames, lasting 2 seconds. We adhere to the official dataset splits, except that the validation split is employed as the test split, following Kipf et al. (2021). Some example frames of the five datasets are shown in Figure 4.

**Metrics.** We assess the quality of 3D slot segmentation masks generated from slot attention masks across the full length of 24 video frames (details in Appendix A.2). Two common metrics for video object learning are used: *FG-ARI* (Greff et al., 2019) and *mIoU*. FG-ARI serves as a clustering similarity metric, measuring the degree to which predicted segmentation masks align with ground-truth masks in a permutation-invariant manner. It only considers foreground pixels, where each cluster corresponds to a foreground segmentation mask. To also evaluate background pixels, mIoU calculates the mean Intersection-over-Union by first employing the Hungarian algorithm to find the optimal bipartite matching (in terms of IoU) between predicted and groundtruth masks, and then dividing sum of the optimal IoUs by the number of groundtruth masks. Both FG-ARI and mIoU demand temporal consistency and penalize object identity switches at any video frame. Nonetheless, neither of them is perfect. For a comprehensive evaluation, a combined interpretation is necessary.

**General training setup.** VONet learns 11 slots on MOVI-{A,B,C} and 16 slots on MOVI-{D,E}. We use a mini-batch size of 32 for MOVI-{A,B,C} and 24 for MOVI-{D,E}. Each sample in a batch has a sequence length of 3. A replay buffer (detailed in Appendix A.2) is employed to enable training from past slot states while preserving the i.i.d. assumption of training data. All video frames are resized to $128 \times 128$. In all subsequent experiments, the learning is entirely unsupervised, meaning that no additional supervision signals, such as depth, optical flow, or segmentation, are provided. For each experiment, we run three random seeds to report the mean and standard deviation of metric

| Method | FG-ARI | | | | | mIoU | | | | |
|--------|--------|--------|--------|--------|--------|--------|--------|--------|--------|--------|
| | MOVI-A | MOVI-B | MOVI-C | MOVI-D | MOVI-E | MOVI-A | MOVI-B | MOVI-C | MOVI-D | MOVI-E |
| SAVI | $45.1_{\pm1.3}$ | $31.8_{\pm1.2}$ | $22.9_{\pm1.3}$ | $29.4_{\pm0.5}$ | $36.0_{\pm3.3}$ | $32.2_{\pm1.7}$ | $31.2_{\pm4.3}$ | $15.9_{\pm0.7}$ | $15.6_{\pm0.3}$ | $15.8_{\pm2.2}$ |
| SIMONe | $50.2_{\pm6.2}$ | $36.5_{\pm0.5}$ | $19.5_{\pm0.1}$ | $34.8_{\pm0.2}$ | $41.3_{\pm0.3}$ | $37.6_{\pm1.0}$ | $35.5_{\pm0.8}$ | $20.2_{\pm0.1}$ | $22.7_{\pm0.1}$ | $20.8_{\pm0.2}$ |
| SCALOR | $68.1_{\pm1.4}$ | $45.3_{\pm0.6}$ | $21.3_{\pm2.3}$ | $33.5_{\pm2.8}$ | $39.6_{\pm0.5}$ | $59.8_{\pm0.9}$ | $46.6_{\pm0.1}$ | $14.3_{\pm0.8}$ | $14.2_{\pm1.1}$ | $12.1_{\pm0.2}$ |
| ViMON | $62.8_{\pm2.3}$ | $26.2_{\pm4.4}$ | $18.0_{\pm2.1}$ | $22.5_{\pm5.7}$ | $17.7_{\pm1.5}$ | $50.0_{\pm1.5}$ | $34.4_{\pm5.0}$ | $27.1_{\pm0.9}$ | $19.6_{\pm1.5}$ | $17.8_{\pm1.2}$ |
| STEVE | $47.8_{\pm8.2}$ | $29.6_{\pm0.3}$ | $38.1_{\pm0.3}$ | $42.9_{\pm2.8}$ | $49.7_{\pm1.0}$ | $53.3_{\pm3.1}$ | $42.7_{\pm0.1}$ | $30.5_{\pm0.2}$ | $23.8_{\pm3.7}$ | $26.2_{\pm1.3}$ |
| VONet | $\mathbf{91.0}_{\pm1.5}$ | $\mathbf{60.6}_{\pm0.8}$ | $\mathbf{45.3}_{\pm0.4}$ | $\mathbf{50.7}_{\pm1.1}$ | $\mathbf{56.3}_{\pm0.5}$ | $\mathbf{60.5}_{\pm2.0}$ | $\mathbf{59.7}_{\pm2.9}$ | $\mathbf{42.8}_{\pm0.5}$ | $\mathbf{37.7}_{\pm0.3}$ | $\mathbf{36.4}_{\pm0.7}$ |

Table 1: Results of VONet and the five baselines, in an unsupervised learning setting.

values. Additional architectural specifics and hyperparameter details are provided in Appendix A.3 for reference. It takes about 36 hours to train VONet on 4x 3090 GPUs on each of the five datasets, for a total number of 150k gradient updates.

## 6.1 EFFICIENCY OF THE PARALLEL ATTENTION

We start with measuring how efficient our parallel attention is compared to the recurrent attention of MONet. With the same U-Net architecture (Appendix A.2), we report the average time of generating attention masks for a varying number of slots on an image of size $128 \times 128$. The results are plotted in Figure 5. VONet demonstrates a strong advantage of being able to maintain a nearly constant inference time regardless of the increasing number of slots. Conversely, MONet's time grows linearly with respect to the slot number, which is expected as MONet generates one attention mask after another. The superior efficiency of VONet on images forms the basis for it being extended to video frame sequences.

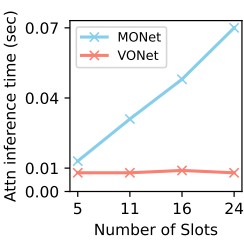

Figure 5: Comparison of the attention inference efficiencies.

## 6.2 COMPARISON WITH BASELINES

**Baselines.** We conduct a comparative analysis of VONet against five representative unsupervised video object learning baselines: SCALOR (Jiang et al., 2019), ViMON (Weis et al., 2021), SIMONe (Kabra et al., 2021), SAVI (Kipf et al., 2021; Elsayed et al., 2022), and STEVE (Singh et al., 2022b). A crucial criterion for baseline selection is the accessibility of an official or public implementation. We employ official implementations of SCALOR, ViMON, SAVI, and STEVE, and a re-implementation of SIMONe (Appendix A.4). SCALOR, ViMON, SIMONe, and STEVE were originally designed for unsupervised learning, with their reconstruction targets being RGB video frames. SAVI, on the other hand, relies on object bounding boxes in initial video frames and reconstructs supervision signals like depth or optical flow. For a fair comparison, we eliminated the bounding box conditioning and only allowed SAVI to reconstruct RGB frames. We adhered to best practices from the literature for configuring the hyperparameters of the five baselines, selecting values recommended either by official codebases or by hyperparameter search results.

**Results.** In Table 1, it is evident that VONet surpasses all the baselines in terms of both FG-ARI and mIoU metrics across all five MOVI datasets. Furthermore, the low std values imply its stability, an important characteristic often lacking in unsupervised video object learning methods. It can also be seen that MOVI-{C,D,E} are indeed much more challenging than MOVI-{A,B}, due to the rich textures and complex backgrounds contained in these datasets. SAVI, SCALOR, and ViMON face challenges when dealing with complex video scenes. Surprisingly, while STEVE performed admirably on MOVI-{C,D,E}, its FG-ARI performance on MOVI-{A,B} was unexpectedly low. Upon closer examination of its visualized results, STEVE tends to over-segment foreground objects and swap object identities, when provided with redundant slots in simpler scenarios (see examples in Figure 7). This behavior significantly diminishes its FG-ARI performance.

## 6.3 HOW CRITICAL ARE DIFFERENT COMPONENTS TO VONET?

We conduct an ablation study to quantify the contribution of various components to the effectiveness of VONet. We select several key ingredients for ablation, resulting in four variants of VONet.

a) *wo-UNet* ablates the U-Net for attention mask generation by removing it, along with the mask transformer at its bottleneck layer, from the parallel attention pipeline. Instead, it directly outputs the estimated attention masks (Section 5). This variant aims to investigate whether the mask refinement provided by the U-Net is indispensable.

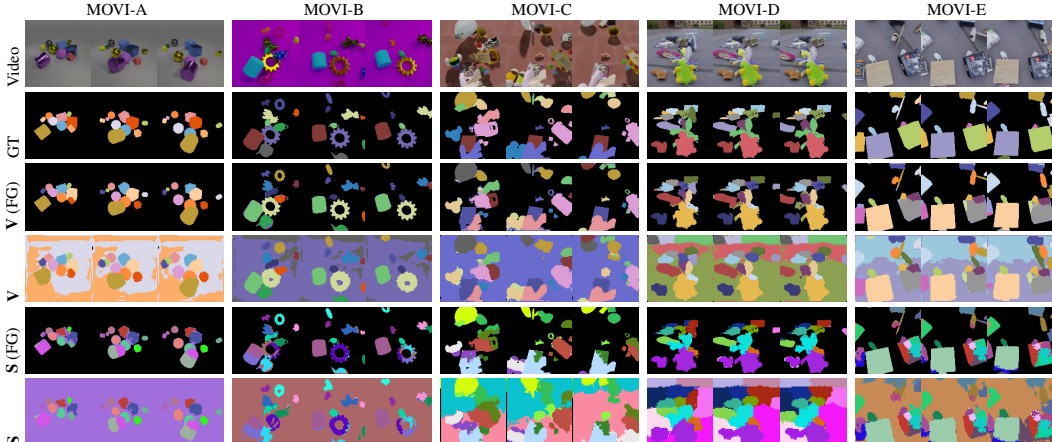

Figure 6: Ablation study results. Each error bar represents the std of three random seeds.

Figure 7: Example segmentation masks of STEVE (**S**) and VONet (**V**). Each column showcases an example video from one of the five datasets. Three key frames are presented in a row for each video. Each unique color represents the mask for a specific slot. **S/V** (FG) shows foreground-only segmentation after background pixels being masked out.

b) *KL-$\frac{\beta}{W}$* reduces the importance of the KLD term in the training loss, where $\beta$ is the KLD weight used by VONet in the experiments. We set $W$ to four values: 20, 200, 20k, and $\infty$, where $\infty$ corresponds to completely eliminating the KLD.

c) *wo-Replay* removes the replay buffer technique and trains from fresh states for each sampled short video segment, as commonly seen in prior works (Kipf et al., 2021; Singh et al., 2022b).

d) *wo-KLBal* excludes the KL balancing technique and calculates a standard KLD loss directly.

All four variants were trained using three random seeds on the five MOVI datasets. For each variant, the remaining training configurations are exactly the same as those used for VONet.

**Results.** Figure 6 shows the impact on the performance if a specific component is removed or weakened in VONet. Notably, without KL balancing, the outcome is very poor on MOVi-A. In this case, VONet struggles to acquire any meaningful mask, as each pixel receives uniform attention from all slots. It is obvious that the posterior has been heavily forced into resembling the prior before the latter is adequately learned. Interestingly, the replay buffer is far more important on MOVI-{A,B} than on the other three. One plausible explanation is that these two datasets can have objects with similar appearances in a video, which makes object tracking more challenging. Training with replayed slot states enhances the long-term object tracking ability of VONet. As for the U-Net architecture, the mask refinement it offers proves to be particularly crucial for MOVI-{C,D,E}. This aligns with our expectation that its inherent spatial locality bias is beneficial for handling complex videos. Lastly, we have observed that the complete removal of the KLD term ($W = \infty$) consistently results in unstable training and eventual crashes, and thus its results were not plotted. Apart from this, we do observe performance declines as the weight decreases, especially on MOVI-{B,C,D,E}. In summary, the four components all prove to be essential. The removal of any of them results in catastrophic outcomes on at least one dataset.

## 6.4 VISUALIZATION AND ANALYSIS

**Object masks.** Figure 7 shows example object masks for VONet and STEVE. While VONet occasionally splits the background, it excels at preserving the structural integrity of moving foreground objects. On the contrary, STEVE tends to over-segment foreground objects, and its object masks exhibit temporal shifts over time. VONet generates smoother and more compact foreground object masks, whereas STEVE produces jagged ones with irregular shapes. We believe that this difference

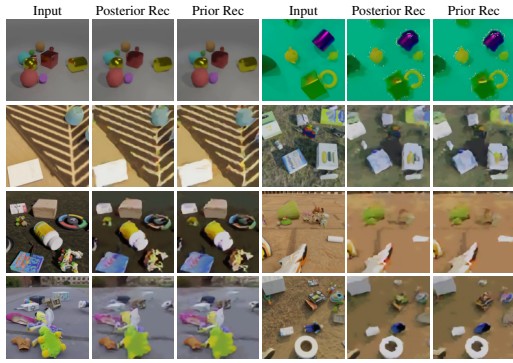 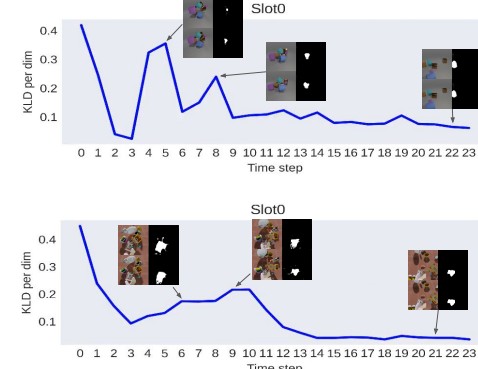

Figure 8: Left: Reconstruction results from the posterior and prior. Right: KLD curves of slot 0 on two example videos. The KLD value has been divided by the slot embedding dimension (128).

arises partially from the strong inductive bias of spatial locality of VONet's U-Net architecture for attention masks. This bias is absent in slot-attention methods like STEVE.

**VAE prior.** One can evaluate the quality of the VAE prior modeling in VONet by reconstructing input frames using forward-predicted slots. We randomly sample multiple video frames and utilize the predicted priors for their reconstruction. The results are shown in Figure 8 (left). We can see that the prior is effectively modeled, since the decoded images generated from the prior closely resembles reconstructed image derived from the posterior slots.

**KLD visualization.** One can also inspect how the per-slot KLD loss in Eq. 8 varies from frame to frame. We anticipate that when a slot exhibits greater temporal variation at specific frames, the corresponding KLD losses will also rise. Figure 8 (right) illustrates two example videos in which the KLD curves are plotted for slot 0. It is evident that the KLD effectively measures the degree of variation in slot content. When the content undergoes significant changes, the KLD loss rises, whereas when the object represented by the slot remains stable, the KLD loss stays at a low level.

**Failure modes.** The first failure mode is over-segmentation (Zimmermann et al., 2023), happening when the video scene's object count is much lower than the pre-allocated slot number. Without extra constraints or priors, VONet aims to use all available slots to encode the video scene for a reduced decoder loss. This causes some objects attended to by multiple slots or the background fragmented into parts (Figure 7, MOVI-A). To address over-segmentation, the model needs to deactivate "redundant" slots. Extra losses may be needed to penalize the use of too many slots. The second failure mode is incomplete object understanding due to the absence of objectness priors. Although we use temporal information for object discovery in videos, the overall learning system remains under-constrained. When an object exhibits multiple texture regions, the model faces challenges in discerning whether it represents a single entity with visually distinct components in motion or multiple distinct objects moving closely together (Figure 7, MOVI-D). Integrating pretrained knowledge about the appearances of everyday objects could help (Zadaianchuk et al., 2023). Lastly, the enforcement of slot temporal consistency may occasionally prove unsuccessful. In certain instances, even when a slot appears to lose temporal consistency, upon closer examination, its KLD loss remains low (Figure 7, the dark-green background slot in MOVI-B), simply because the VAE prior is accurately predicted. This suggests an opportunity for improving our KLD loss. The temporal consistency might also benefit from using a long-term memory model (Cheng & Schwing, 2022) as opposed to the current short-term GRU memory which might get expired under long-time occlusion.

## 7 CONCLUSION

We have presented VONet, a state-of-the-art approach for unsupervised video object learning. VONet incorporates a parallel attention process that simultaneously generates attention masks for all slots from a U-Net. The strong inductive bias of spatial locality in the U-Net leads to smoother and more compact object segmentation masks, compared to those derived from slot attention. Furthermore, VONet effectively tackles the challenge of temporal consistency in video object learning by propagating context vectors across time and adopting an object-wise sequential VAE framework. Across five datasets comprising videos of diverse complexities, VONet consistently demonstrates superior effectiveness compared to several strong baselines in generating high-quality object representations. We hope that our findings will offer valuable insights for future research in this field.

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

| Method | Image FG-ARI | | | Image mIoU | | |
|---|---|---|---|---|---|---|
| | MOVI-C | MOVI-D | MOVI-E | MOVI-C | MOVI-D | MOVI-E |
| MONet | $24.8_{\pm 0.1}$ | $22.0_{\pm 1.7}$ | $18.8_{\pm 0.9}$ | $34.8_{\pm 0.9}$ | $17.4_{\pm 2.0}$ | $13.6_{\pm 1.6}$ |
| VONet | $47.0_{\pm 2.8}$ | $51.0_{\pm 1.2}$ | $50.3_{\pm 3.6}$ | $36.2_{\pm 3.0}$ | $28.5_{\pm 0.6}$ | $18.1_{\pm 1.3}$ |

Table 2: Results of VONet and MONet in the single-frame scenario.

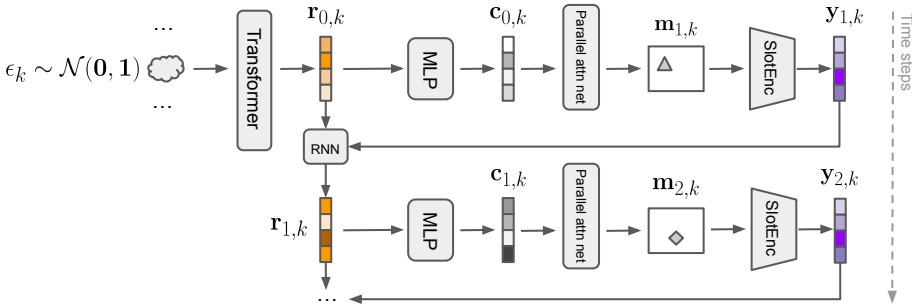

Figure 9: Illustration of the unrolling process for VONet on a video sequence. Only slot $k$ is depicted in the diagram, with the other slots following similar flows. Slot computations can be parallelized within each step, except when inter-slot interaction is required (*e.g.* initial slot transformer and parallel attention). For simplicity, we omit depicting the dependencies on the input frames $\mathbf{x}_t$.

## A    APPENDIX

### A.1    SINGLE-FRAME SCENARIO

We compared VONet with MONet in the single-frame scenario, as MONet was designed for object discovery in individual images only. We configured the U-Net architecture of MONet to match that of VONet, and set MONet's $\beta = 1$ and $\gamma = 2$ after a brief exploration within a reasonable parameter range. VONet was modified to train with a sequence length of 1 and excluded training from past slot states. For the training and evaluation data, we utilized video frames from MOVI-{C,D,E}, treating them as independent images with no temporal relationship. FG-ARI and mIoU are calculated on individual frames, referred to as Image FG-ARI and Image mIoU. The results are shown in Table 2. They suggest that VONet is not only more efficient than MONet, but also more competent to discover objects from complex images.

### A.2    MORE DETAILS OF VONET

**Backbone for image features.** We train a backbone based on a small ResNet (He et al., 2016) to extract a feature map from each input frame $\mathbf{x}_t$. The backbone architecture includes five residual blocks, each employing a $3 \times 3$ kernel and a stride of 1, except for the first block which has a stride of 2. All the blocks have an output channel number of 64. Following the ResNet, there is a concluding convolution layer with a $1 \times 1$ kernel and a linear activation, with an output channel number of 128. Finally, a position embedding network (Kipf et al., 2021) is applied to provide 2D location information. With this backbone, the output feature map has a spatial shape that is half of that of the input frame. This feature map is shared by the parallel attention network and the slot encoder as the input.

**U-Net and mask transformer.** The U-Net used by the parallel attention network follows a standard architecture closely resembling that of MONet (Burgess et al., 2019). There are $M$ blocks on either the upsampling or downsampling path, with $M$ taking the value of 5 for datasets MOVI-{A,B} and 6 for datasets MOVI-{C,D,E}. Each block comprises the following layers in order: (a) a $3 \times 3$ bias-free convolution with a stride of 1 and a padding of 1, (b) an instance normalization (Ulyanov et al., 2016) layer with learnable affine parameters, (c) a ReLU activation, and finally (d) a max pooling layer of size 2 for the downsampling path and a nearest upsampling layer of factor 2 for the upsampling path. No max pooling (upsampling) is applied to the first (last) block of the downsampling (upsampling) path. A skip connection is established between the output of the $m$-th downsampling

block and the output of the $(M - m + 1)$-th upsampling block, for each $2 \leq m \leq M$. The non-skip bottleneck connection is an MLP with two layers of sizes $(512, 512)$, both activated with ReLU. The output of this MLP will be projected and reshaped to match the output of the downsampling path, concatenated with it, and input to the upsampling path. After the upsampling path, a $1 \times 1$ convolution is applied with an output channels of 1. The final output is a 2D map that retains the same spatial dimensions as the input. The values within this map represent unnormalized attention logits. For the downsampling blocks, we set their output channels as $\{32, 64, 64, 128, 128\}$ for MOVI-$\{A,B\}$, with an additional output channel number of 128 for MOVI-$\{C,D,E\}$.

To prepare the input to the U-Net for a slot's attention mask, we first convolve its context vector across the backbone feature locations, yielding a roughly estimated attention mask for that slot. Then its context vector is spatially broadcast to the feature locations, forming a location-invariant context map. Finally, the backbone feature map, the estimated attention mask, and the context map, are concatenated in the channel dimension to form the input to the U-Net.

When there are $K$ slots whose attention masks are being computed in parallel, right after the U-Net bottleneck MLP, we use a mask transformer to model the interaction among the $K$ masks in the latent space. The transformer consists of three decoder-only transformer blocks, and each block can be briefly described as:

$$\mathbf{u}' = \mathbf{v} + \text{MLP}(\text{LayerNorm}(\mathbf{v})), \quad \mathbf{v} = \mathbf{u} + \text{MultiHeadAttn}(\text{LayerNorm}(\mathbf{u})).$$

where $\mathbf{u}$ represents the $K$ mask latent vectors and $\mathbf{u}'$ are the updated latent vectors. We configure each block with three attention heads. To maintain permutation invariance for the slots, no position encoding is incorporated in any of the transformer blocks.

**Initial slot latent.** The initial slot latent transformer in Eq. 5 shares a similar architecture with the U-Net mask transformer, differing only in terms of layer sizes. It also has three decoder-only transformer blocks and three attention heads per block. We set both the per-trajectory slot latent size and the per-frame slot latent size to be 128.

**VAE prior and posterior.** To compute the prior, again we use a transformer (Eq. 7) that has a similar architecture with the U-Net mask transformer, except for different layer sizes. The transformer has only two blocks with three attention heads in each block. Its output $\mathbf{r}'_{t,k}$ is independently projected (via an MLP of one hidden layer of size 128) to generate the mean and log variance of a Gaussian as the prior distribution. The posterior is obtained by projecting the updated per-trajectory slot latent $\mathbf{r}_{t,k}$ to generate its mean and log variance with a similar MLP.

**Decoder.** Singh et al. (2022a) identified the mixture of components decoder in MONet as a weak decoder, pinpointing two primary limitations: the slot-decoding dilemma and pixel independence. These limitations pose challenges to the encoder's capacity to achieve effective scene decomposition. As a remedy, they proposed using a more powerful transformer-based decoder (Chen et al., 2020) instead. This decoder attends to all slots simultaneously and decodes the image in an autoregressive manner. At a high level, the decoder is formulated as:

$$P_{\text{TransDec}}(\mathbf{x}|\mathbf{z}_1, \ldots, \mathbf{z}_K) = \prod_m P(\mathbf{x}(m)|\mathbf{z}_1, \ldots, \mathbf{z}_K, \mathbf{x}(1), \ldots, \mathbf{x}(m-1)). \tag{11}$$

Here, $\mathbf{x}(m)$ represents the $m$-th patch on the image in the row-major order. Singh et al. (2022b) applied this decoder to learning objects from complex videos and achieved a notable performance enhancement.

We directly reused the transformer decoder implementation from the official code[2] of STEVE (Singh et al., 2022b). For the discrete VAE model, we re-implemented our own, but closely following the official implementation. All the decoder-related hyperparameters used in our experiments kept the same with those of STEVE.

**Segmentation mask.** For FG-ARI and mIoU evaluation, an integer segmentation mask is generated by consolidating the $K$ slot attention masks $\mathbf{m}_{t,k}$ (Eq. 3), where each mask contains a floating-point value within the range of $[0, 1]$ at each pixel location. This value represents the probability of the pixel being assigned to the corresponding slot. If the maximum probability at the pixel is smaller than $0.3$ (no slot is confident enough), we assign that pixel to a special "null" slot. Otherwise, the

---

[2]https://github.com/singhgautam/steve

pixel is assigned to the slot with the maximum probability. As a result, each segmentation mask contains at most $K + 1$ distinct integer values, where $K$ is the number of pre-allocated slots.

**Training from replayed video segments.** Training VONet on entire videos is impractical due to high memory usage caused by the large computational graph unrolled over time. Traditional video object learning methods, like Jiang et al. (2019); Kipf et al. (2021); Singh et al. (2022b), train models on short video segments, assuming that each mini-batch is initialized with fresh slots. During testing, they expect models to generalize to longer videos. VONet is also trained on short video segments, but it stores past states in a replay buffer, where each state includes per-trajectory slot latents $\{\mathbf{r}_{t,k}\}_{k=1}^{K}$ and a boolean flag indicating the initial frame of a video. When sampling a video segment from the replay buffer, we initiate the training of VONet with the state of the first step of that segment, and only reset the state if the flag is true. Despite potential state obsolescence, this replay-based training approach proves effective if a small replay buffer length is used. Using a replay buffer allows training short video segments from past slot states, while still preserving the i.i.d. assumption with a random replay strategy. Without it, online training from previous slot states requires sequential training through entire videos, introducing temporal correlation in the training data.

During training, we create a replay buffer with a length of 10k frames and a width of 16 videos. For each gradient update, we first unroll the same copy of VONet on each of 16 videos for 2 steps, store the unrolled states in the replay buffer, and randomly extract a mini-batch of size $B$ with a length of 3 (in total $3B$ frames) from the buffer to compute the gradient. After the update, the unrolling continues until the videos are finished (after $\frac{24}{2} = 12$ updates), when we randomly select another 16 videos from the dataset to replace them. We set $B = 32$ for MOVI-{A,B,C} while $B = 24$ for MOVI-{D,E}.

## A.3 HYPERPARAMETERS

We provide a brief description of the key hyperparameters used for VONet in our experiments. For the complete set of hyperparameters, we encourage the reader to refer to our code [3] .

The input video frame $\mathbf{x}_t$ is resized to $128 \times 128$. Unlike some prior methods, we did not perform any data augmentation on the frames. We set the lengths of all the following vectors to 128:

  i) Noise vector $\epsilon_k$ (Eq. 5);
 ii) Per-frame slot latent $\mathbf{y}_{t,k}$ (Eq. 4);
iii) Per-trajectory slot latent $\mathbf{r}_{t,k}$ (Eq. 4);
 iv) Context vector $\mathbf{c}_{t,k}$ (Eq. 4);
  v) VAE posterior embedding $\mathbf{z}_{t,k}$ (Eq. 6).

(We also explored a smaller length of 64 for these vectors, but obtained slightly worse results.) For any transformer, its model size (`d_model`), key size (`d_k`), and value size (`d_v`) are all set to be equal to the query size, whereas its hidden size (`d_ff`) is twice the query size.

We rolled out 16 workers in parallel for collecting video data in the replay buffer. Each worker utilizes the most recent VONet model for inference on 2 consecutive video frames, stores the inputs and states in the replay buffer, pauses for the trainer to perform a gradient update, and then resumes data collection for another 2 frames, following this iterative process. The replay buffer length was set to 10k, resulting in a maximum number of time steps $16 \times 10k = 160k$ in the buffer.

We trained 11 slots for MOVI-{A,B,C} while 16 slots for MOVI-{D,E}, reflecting the increased maximum number of objects in the latter two datasets. Accordingly, to ensure consistent GPU memory consumption across all training runs, we employed a mini-batch size[4] of 32 for MOVI-{A,B,C} and 24 for MOVI-{D,E}. We sampled video segments of length 3 for training on all datasets. This training configuration ensures that any training job can be executed on a system equipped with 4 NVIDIA GeForce RTX 3090 GPUs or an equivalent amount of GPU memory.

For the optimization, we set the KL balancing coefficient $\kappa$ (Eq. 10) to 0.7. The KL loss weight $\beta$ (Eq. 8) was linearly increased from 0 to $\frac{20}{D}$ in the first 50k updates, where $D = 128$ is the posterior

---

[3] https://github.com/hnyu/vonet
[4] In our paper, when we refer to a sample within a mini-batch, we are referring to a video segment. Therefore, when we specify a mini-batch size of $N$, it indicates that the batch consists of $N$ video segments.

dimension. We used the Adam optimizer (Kingma & Ba, 2014) with a learning rate schedule as follows. The learning rate was increased linearly from $10^{-5}$ to $10^{-4}$ in the first 5k training updates, maintained the value until 100k updates, and finally was decreased linearly back to $10^{-5}$ in another 50k updates. Thus in total, we trained each job for 150k gradient updates. In each update step, we normalized the global parameter norm to 0.1 if the norm exceeds this threshold.

## A.4 BASELINE DETAILS

All baseline methods except SIMONe (reason explained below), were trained with 11 slots for MOVI-{A,B,C} and 16 slots for MOVI-{C,D}. All baseline methods, with the exception of SCALOR, employed the same methodology for obtaining segmentation masks as was utilized in VONet. SCALOR, on the other hand, possesses its own robust method for deriving segmentation masks from attention maps, and thus we did not replace it. Below are their detailed training settings.

**SCALOR.** We used the official implementation of SCALOR at https://github.com/JindongJiang/SCALOR. We followed the suggestions in SAVI (Kipf et al., 2021) to configure its hyperparameters, with several exceptions as follows. The training video segment length was reduced from 10 to 6 to reduce memory consumption. We used an $8 \times 8$ grid instead of the $4 \times 4$ grid in SAVI as we found the former produced much better results on MOVI-{A,B}. We used a learning rate of $10^{-4}$ to speed up the training convergence.

**ViMON.** We used the official implementation of ViMON at https://github.com/ecker-lab/object-centric-representation-benchmark. We made adjustments to its default hyperparameters to adapt its training to the MOVI datasets, by setting the VAE latent dim to 64 and the training video seq length to 6. We also explored different values for $\gamma$, but found the default value 2 is already good enough. All other hyperparameters remain unchanged.

**SAVI.** We made several changes to the official implementation of SAVI at https://github.com/google-research/slot-attention-video for our experiments. First, SAVI was trained in an entirely unsupervised manner to reconstruct RGB video frames only, without the object bounding box conditioning in the first video frame. The slots were initialized as unit Gaussian noises as in VONet. Second, to mitigate GPU memory usage, we halved the original training batch size, reducing it from 64 to 32. We also reduced the training video segment length from 6 to 3 on MOVI-{D,E}. Remarkably, these size adjustments did not yield any discernible impact on the training results according to our observations on several trial training jobs. Finally, we conducted training with a medium-sized SAVI model, featuring a 9-layer ResNet as the encoder backbone which has a comparable size to the backbone of VONet. Specifically, the ResNet architecture was created by assigning a size (number of residual blocks) of 1 to each of the four ResNet stages, employing the `class` `ResNetWithBasicBlk` implemented by SAVI.

**SIMONe.** We used the re-implementation of SIMONe at https://gitlab.com/generally-intelligent/simone, which successfully reproduced the performance on CATER (Girdhar & Ramanan, 2019). In contrast to other baseline methods and VONet, SIMONe imposes a strict requirement on the number of learnable slots. This number must be equal to the product of the height and width of the feature map resulting from its CNN encoder and transformer. Consequently, we employed a fixed slot number of 16 across all MOVI datasets for SIMONe. We set the mini-batch size to 40 and the video segment length to 24 (full length). All other hyperparameters remained unaltered.

**STEVE.** Since STEVE has been extensively evaluated on MOVI-{D,E} by its authors, we largely retained its original hyperparameters. However, we explored one exception: experimenting with two different values for its slot size: 64 and 128. Our observation revealed that a larger slot size of 128 consistently yielded no better or even sometimes slightly inferior results across all five datasets. Consequently, we settled on a slot size of 64 for STEVE in our final configuration.

## A.5 DATASET DETAILS

We refer the reader to the official MOVI datasets website for details: https://github.com/google-research/kubric/tree/main/challenges/movi. We did not make any change to the five datasets. The official training/validation splits were used in our experiments.

## A.6    MORE VISUALIZATION RESULTS

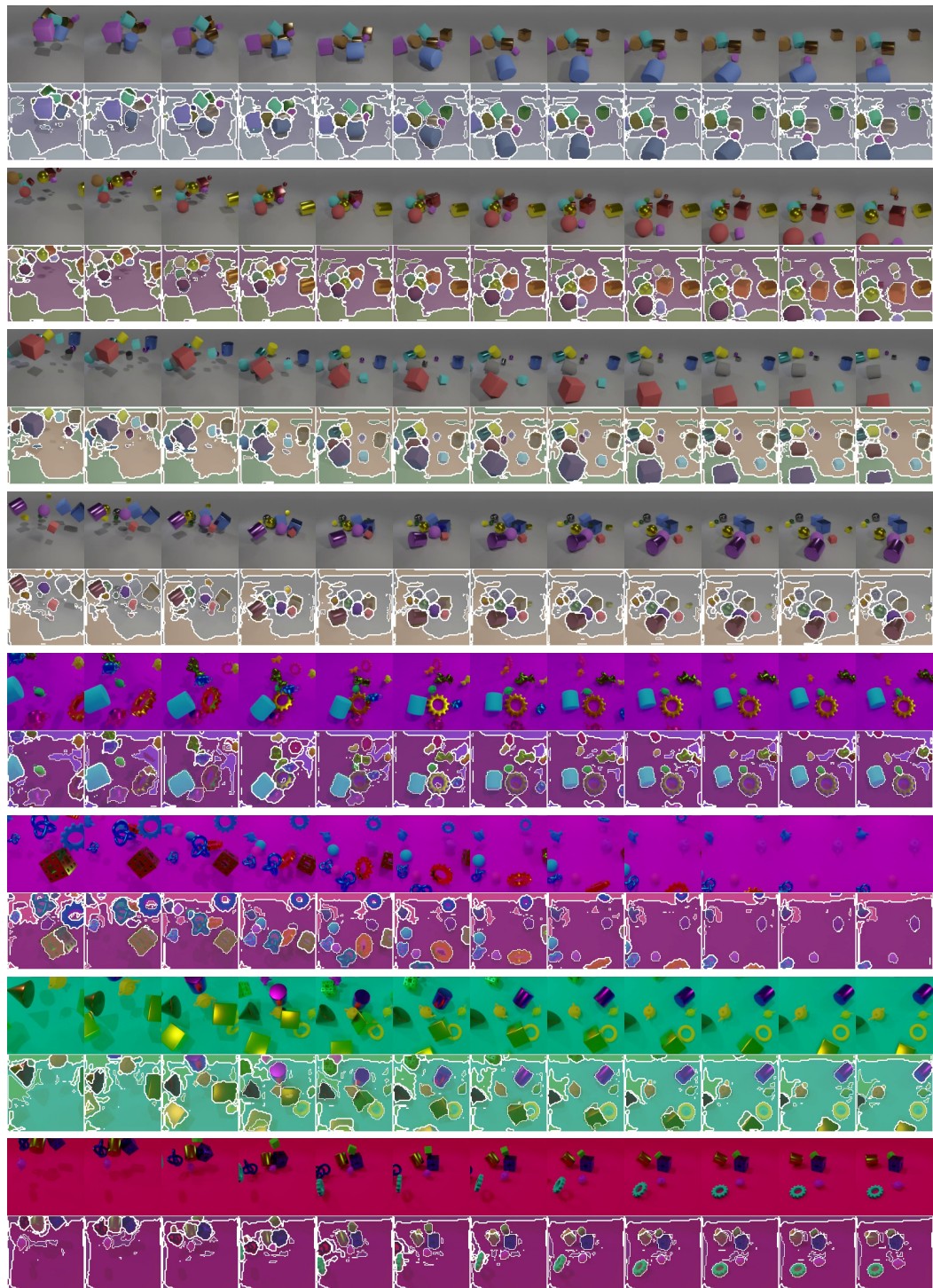

Figure 10: Additional segmentation results of VONet. In these results, each video is presented with every other frame displayed. The boundaries of object segments are marked with white curves. No post-processing was performed on the masks.

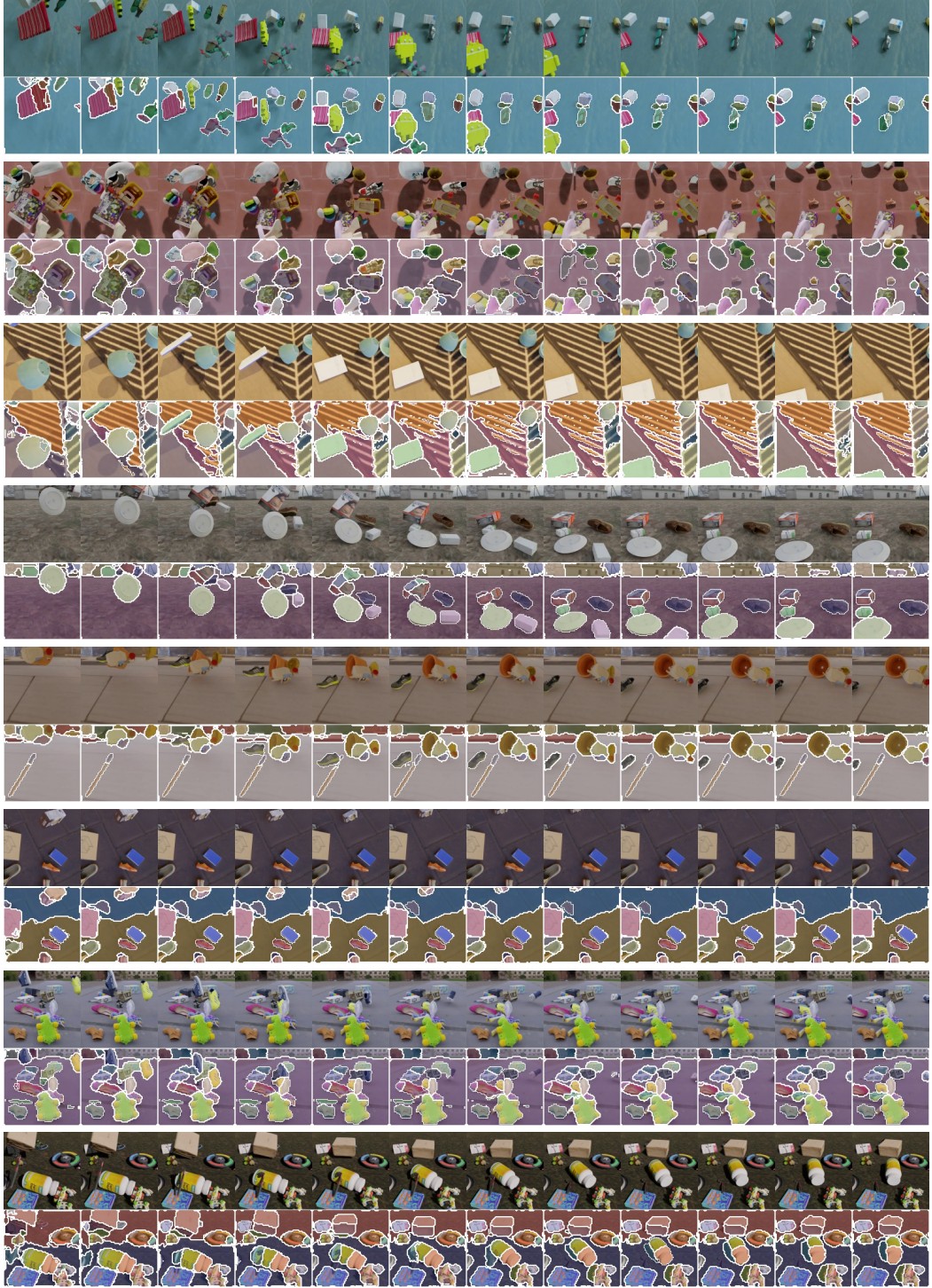

Figure 11: Additional segmentation results of VONet. In these results, each video is presented with every other frame displayed. The boundaries of object segments are marked with white curves. No post-processing was performed on the masks.

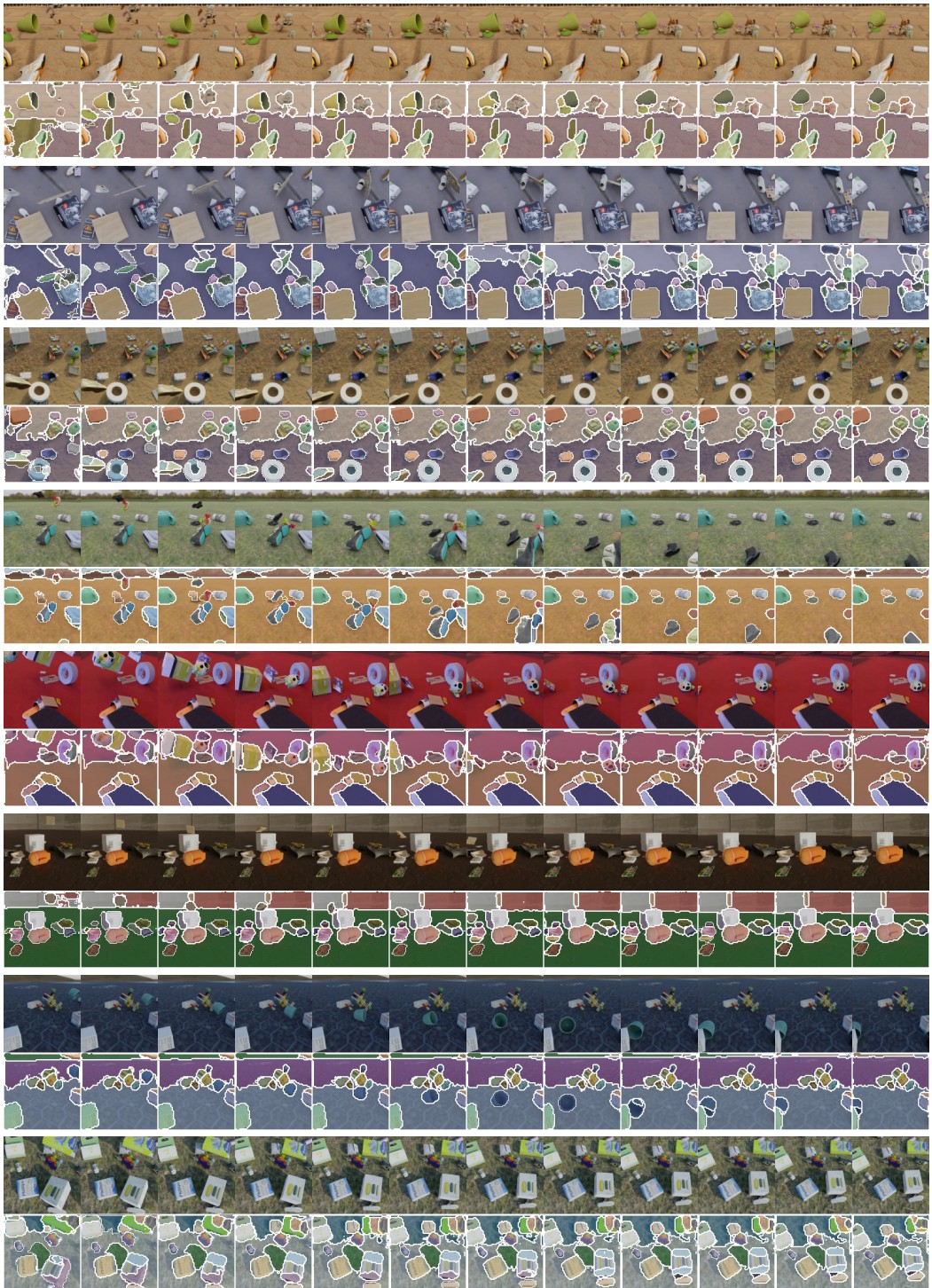

Figure 12: Additional segmentation results of VONet. In these results, each video is presented with every other frame displayed. The boundaries of object segments are marked with white curves. No post-processing was performed on the masks.

