# OpenReview forum: "VONet: Unsupervised Video Object Learning With Parallel U-Net Attention and Object-wise Sequential VAE"
_ICLR.cc/2024/Conference — ICLR 2024 poster_

### Official Review · Reviewer_751B · 2023-10-28

**Soundness:** 2 fair
**Presentation:** 3 good
**Contribution:** 2 fair
**Rating:** 5
**Confidence:** 4

**Summary:**

This paper proposes an unsupervised video object representation learning framework, namely VONet. Taking an image-based method as per-frame baseline, the proposed VONet builds temporal attention to learn correspondence in high-level space, resulting in significant improvement against previous video-based methods.

**Strengths:**

1. The results look good with large improvement against previous methods.

**Weaknesses:**

1. The current title is too broad, which makes the readers hard to understand the specific contributions of this paper.

2. The motivation of each contribution of the proposed method is not clearly clarified in Introduction, especially the difference or new insights w.r.t previous  methods.

3. It is hard to refer to the expression of the symbols in Figure 3.

**Questions:**

See weakness.

---

> ### Author Response · Authors · 2023-11-21
>
> > The current title is too broad, which makes the readers hard to understand the specific contributions of this paper.
>
> We have updated the title to make it more specific:
>
> "VONet:  Unsupervised Video Object Learning With Parallel U-Net Attention and Object-wise Sequential VAE",
>
> to highlight the two key innovations of the paper, as appreciated by most reviewers.
>
> > The motivation of each contribution of the proposed method is not clearly clarified in Introduction, especially the difference or new insights w.r.t previous methods.
>
> The second paragraph in the Introduction section talks about the motivation and contributions of VONet. We agree that the current paragraph might be a little short. We've expanded the Introduction section to elaborate the motivation and contributions of this work. Please also see our consolidated response for more clarifications.
>
> > It is hard to refer to the expression of the symbols in Figure 3.
>
> We've updated Figure 3 by adding a text label besides each symbol, to provide a better context. Also we'd like to clarify that the '*' in each symbol's subscript represents the collection of $K$ slots in parallel (only two slots are plotted in the figure). The updated figure is below:
>
> https://i.imgur.com/UijSdnc.png

---

### Official Review · Reviewer_x2B3 · 2023-10-30

**Soundness:** 3 good
**Presentation:** 3 good
**Contribution:** 3 good
**Rating:** 6
**Confidence:** 3

**Summary:**

The paper proposes a framework for unsupervised video object learning, VONet. The uniqueness of VONet is primarily the parallel attention process that is capable of generating attention masks for all slots with the consideration of temporal consistency which utilizes context propagation across time and object-wise sequential VAE framework. Their results on 5 MOVI datasets show that the proposed method significantly outperforms previous methods as measured by two popular metrics namely FG-ARI and mIoU.

**Strengths:**

The proposed method addresses unsupervised video object learning which can be paralleled and temporally consistent given the slot numbers. Both temporal consistency and parallel segmentation instead of a sequential learning are crucial for learning objects in a video. This paper solves these very important first steps.

**Weaknesses:**

As already mentioned in the paper when the predefined slot numbers is larger than the actual number of objects, an unwanted side-effect if over-segmentation. Can the authors provide any insights on how to potentially combine multiple slots to prevent such overfitting?

**Questions:**

Can the authors provide any insights on how to potentially combine multiple slots to prevent such overfitting?

---

> ### Author Response · Authors · 2023-11-21
> **Response to Reviewer x2B3**
>
> > As already mentioned in the paper when the predefined slot numbers is larger than the actual number of objects, an unwanted side-effect if over-segmentation. Can the authors provide any insights on how to potentially combine multiple slots to prevent such overfitting?
>
> Instead of combining multiple slots into one, a possible alternative is to let the attention process learns to disable a certain slot by outputting a very small probability of that slot over *all pixels*. Imagine that in addition to generating a slot probability distribution at every pixel location, the attention network also generates a global gate signal to control the on/off of that slot.  This gate value in $[-\infty,0]$, same over all pixels on an image, will be added to the attention logits of that slot at every pixel. If the gate is very negative, then that slot is basically turned off.
>
> Besides this architecture, we also need some objective to drive the model to learn such gate values in the way we want. In general, when the number of objects is small in a video, we don't want to waste slots to encode and oversegment an object. If we treat slots as valuable resources, then an objective that penalizes turning on slots will be helpful (e.g, penalizing gate values close to 0). There is inherently a conflict between reconstructing the input image (better if more slots are used) and turning on fewer slots. How to cleverly strike a balance between the two is an interesting challenge.

---

### Official Review · Reviewer_9HJm · 2023-11-04

**Soundness:** 3 good
**Presentation:** 3 good
**Contribution:** 3 good
**Rating:** 6
**Confidence:** 2

**Summary:**

VONet introduces two key innovations: 1) a parallel attention network that employs the same U-Net architecture simultaneously on K context inputs, and 2) an object-wise sequential VAE framework, aimed at improving the temporal consistency in unsupervised video object learning. Notably, VONet significantly outperforms the baseline by a substantial margin.

**Strengths:**

1. Within this paper, the "object-wise sequential VAE" is introduced, which is a novel and highly effective representation for exploring temporal dependencies in video frames.
2. The experimental results in this paper are impressive, surpassing previous methods by a large margin.

**Weaknesses:**

1. In order to show the advantages of parallel processing, it would be beneficial that a comprehensive latency/accuracy comparison with MONet could be provided in the single-frame scenario.

2. It would be beneficial to have a comparative analysis between the object-wise sequential VAE and other temporal dependency networks, particularly the memory network mentioned in reference [1]. These methods excel in modeling temporal information and such a comparison would greatly enhance the paper's overall comprehensiveness.

[1] XMem: Long-Term Video Object Segmentation with an Atkinson-Shiffrin Memory Model ECCV 2022

**Questions:**

Please see Weaknesses section.

---

> ### Author Response · Authors · 2023-11-21
> **Response to Reviewer 9HJm**
>
> > In order to show the advantages of parallel processing, it would be beneficial that a comprehensive latency/accuracy comparison with MONet could be provided in the single-frame scenario.
>
> **Latency comparison**. With the same U-Net architecture, we report the average time of generating attention masks for a varying number of slots on an image of size 128x128. With a mini-batch size of 32, we timed the attention inference process for a total of 200 calls on a 3090 GPU, and recorded the average time. The results are summarized in the figure below. VONet demonstrates a strong advantage of being able to maintain a nearly constant inference time regardless of the increasing number of slots. In stark contrast, MONet’s time grows linearly with respect to the slot number, which is expected as MONet generates one attention mask after another. The superior efficiency of VONet on individual frames forms the basis for extending it to video frame sequences.
>
> https://i.imgur.com/5Zt4KEJ.png
>
> **Accuracy comparison**. We compare VONet with MONet on individual video frames from MOVI-{C,D,E}. We modified VONet to train with a sequence length of 1 and excluded training from past slot states. FG-ARI and mIoU are calculated on individual frames, referred to as Image FG-ARI and Image mIoU.
>
> | **Method** | **Image FG-ARI (MOVI-C)** | **Image FG-ARI (MOVI-D)** | **Image FG-ARI (MOVI-E)** | **Image mIoU (MOVI-C)** | **Image mIoU (MOVI-D)** | **Image mIoU (MOVI-E)** |
> |------------|---------------------------|---------------------------|---------------------------|-------------------------|-------------------------|-------------------------|
> | MONet      | $24.8{\scriptstyle \pm 0.1}$ | $22.0{\scriptstyle \pm 1.7}$ | $18.8{\scriptstyle \pm 0.9}$ | $34.8{\scriptstyle \pm 0.9}$ | $17.4{\scriptstyle \pm 2.0}$ | $13.6{\scriptstyle \pm 1.6}$ |
> | VONet  | $47.0{\scriptstyle \pm 2.8}$ | $51.0{\scriptstyle \pm 1.2}$ | $50.3{\scriptstyle \pm 3.6}$ | $36.2{\scriptstyle \pm 3.0}$ | $28.5{\scriptstyle \pm 0.6}$ | $18.1{\scriptstyle \pm 1.3}$ |
>
>
> The results suggest that VONet is not only more efficient than MONet, but also more competent to discover objects from complex images. We've put this experiment in the Appendix due to page limit.
>
> > It would be beneficial to have a comparative analysis between the object-wise sequential VAE and other temporal dependency networks, particularly the memory network mentioned in reference [1]. These methods excel in modeling temporal information and such a comparison would greatly enhance the paper's overall comprehensiveness.
>
> Thanks for providing the reference to XMem [Cheng and Schwing 2022]. XMem suggests three memory types for tracking a single object in a lengthy video: a quickly updated sensory memory, a detailed working memory, and a compact, sustained long-term memory. It also introduces a technique to transfer information among these memory stores. The primary focus of XMem is addressing the challenge of maintaining an efficient long-term memory representation of an object over extended video durations without suffering from memory explosion.
>
> Our object-wise sequential VAE relies on a GRU to maintain the slot history. In this sense, we only exploit the first memory type of XMem: the sensory memory. This choice aligns with our primary focus on ensuring temporal consistency/smoothness of an object between adjacent frames. This model might indeed fail to track an object if it is out of the scene or occluded for a very long time and re-appears in the video, as the GRU state of a slot could expire or be wiped out as time proceeds. Thus our object-wise sequential VAE would definitely benefit from a long term memory such as XMem for solving the long-term occlusion problem.
>
> However, the object-wise sequential VAE also emphasizes modeling the dynamic interaction and coevolvement of multiple objects within the scene as a significant focal point. The predictive prior in our model is generated by employing a transformer on the latents of the $K$ slots in the preceding time step (refer to Eq 7). Minimizing the KLD between the posterior and the forecasted prior ensures temporal consistency of the slots in a holistic manner. This distinctive feature is absent in XMem, as it pursues a different objective.
>
> In summary, our VAE framework targets short-term temporal consistency with a focus on multi-object interaction. A promising avenue for further exploration involves integrating a long-term slot memory representation into VONet. We've added XMem [Cheng and Schwing 2022] as a reference in the paper.

---

> > ### Comment · Reviewer_9HJm · 2023-11-23
> >
> > Thanks for the rebuttal. It addressed me concerns and I would like to keep me original score.

---

### Official Review · Reviewer_v4jo · 2023-11-04

**Soundness:** 3 good
**Presentation:** 2 fair
**Contribution:** 2 fair
**Rating:** 5
**Confidence:** 4

**Summary:**

This paper presents VONet, an innovative approach to unsupervised video object learning inspired by Monet. Specifically, Monet implements an efficient and effective parallel attention inference process that simultaneously generates attention masks from U-Net for all slots. In addition, the temporal consistency necessary to track objects across video frames is achieved by integrating an object-wise sequential VAE framework. Experiments demonstrate that the approach achieves competitive performance on several challenging object-centric video prediction benchmarks.

**Strengths:**

1.The parallel attention inference process proposed in this paper greatly improves the slot generation efficiency of MoNet and creates conditions for its further application.

2.This paper proposes the KLD loss, which utilizes the principle that "only slot representations that exhibit temporal consistency can exhibit predictability" to cleverly achieve temporal consistency in unsupervised video object learning.

3.The paper is relatively easy to follow, with good mathematical formulations and diagrams.

**Weaknesses:**

1.VoNet is a work based on MoNet and is highly similar to ViMON, a comparison of results with MoNet and ViMON should be reported in the experimental phase to demonstrate the advantages of VoNet.

2.Can the method in this paper correctly handle objects appearing in the middle of the video or reappearing after being occluded and maintain temporal consistency? The MOVI dataset does not seem to be able to model this situation, consider adding experiments in natural scenes.

3.Existing video object-centric learning methods based on slot attention are developing rapidly, e.g.DINOSAUR(https://arxiv.org/abs/2209.14860), VideoSAUR(https://arxiv.org/pdf/2306.04829.pdf), and perform well on natural scene datasets such as YouTube-VIS, please explain the advantages of the method in this paper compared to methods based on slot attention.

**Questions:**

For writng, the introduction section is short. It is hard to understand the main idea of the whole work.

---

> ### Author Response · Authors · 2023-11-21
> **Response to Reviewer v4jo**
>
> >  VoNet is a work based on MoNet and is highly similar to ViMON, a comparison of results with MoNet and ViMON should be reported in the experimental phase to demonstrate the advantages of VoNet.
>
> Although both inspired by MONet, VONet and ViMON are different in many ways, as stated in Section 2 (last paragraph):
>
> "*...  several major differences exist between the two, including: i) VONet parallelizes the original MONet architecture for efficient inference while ViMON still generates slots recurrently. ii) VONet seeds the attention network with context vectors while ViMON does this using previous attention masks. iii) VONet formulates an object-wise sequential VAE to promote temporal consistency, whereas ViMON ignores this by using a typical intra-step VAE..*."
>
> Regarding i), a parallel attention process can greatly speed up the inference, especially when the number of slots increases. Please see our additional experiment results in response to Reviewer 9HJm on the latency differences between MONet and VONet on single frames. On a single GPU, VONet's attention inference time remains constant w.r.t. the number of slots, while MONet's inference time grows linearly. The superior efficiency of VONet in the single-frame scenarios forms the basis for its practical application to video sequences. As for ii), note that when generating attention, our context vector can in theory encode information about the history of a slot all the way back to T=0, while VIMON conditions only on the attention masks of T=t-1 . This property enables VONet to handle occlusions (see our response to the next question). Lastly for iii), our object-wise sequential VAE is a crucial component to ensure temporal consistency as verified by our ablation study (Figure 6), and such a similar formulation is lacked in VIMON.
>
> As suggested by the reviewer, we have added ViMON as another baseline to our experiments (Section 6.2). We used the official released code of ViMON (https://github.com/ecker-lab/object-centric-representation-benchmark) and made several adjustments to its default hyperparameters to adapt its training to the MOVI datasets. We set the VAE `latent_dim=64`, `n_steps=6` (training video seq length), and updated `n_slots` to either 11 (MOVI-{A,B,C)) or 16 (MOVI-{D,E}) as in other comparison methods. The results are below:
>
> | **Method** | **FG-ARI (MOVI-A)** | **FG-ARI (MOVI-B)** | **FG-ARI (MOVI-C)** | **FG-ARI (MOVI-D)** | **FG-ARI (MOVI-E)** | **mIoU (MOVI-A)** | **mIoU (MOVI-B)** | **mIoU (MOVI-C)** | **mIoU (MOVI-D)** | **mIoU (MOVI-E)** |
>  |------------|----------------------|----------------------|----------------------|----------------------|----------------------|---------------------|---------------------|---------------------|---------------------|---------------------|
>  | ViMON | $62.8{\scriptstyle \pm 2.3}$ | $26.2{\scriptstyle \pm 4.4}$ | $18.0{\scriptstyle \pm 2.1}$ | $22.5{\scriptstyle \pm 5.7}$ | $17.7{\scriptstyle \pm 1.5}$ | $50.0{\scriptstyle \pm 1.5}$ | $34.4{\scriptstyle \pm 5.0}$ | $27.1{\scriptstyle \pm 0.9}$ | $19.6{\scriptstyle \pm 1.5}$ | $17.8{\scriptstyle \pm 1.2}$ |
>  | VONet | $\mathbf{91.0{\scriptstyle \pm 1.5}}$ | $\mathbf{60.6{\scriptstyle \pm 0.8}}$ | $\mathbf{45.3{\scriptstyle \pm 0.4}}$ | $\mathbf{50.7{\scriptstyle \pm 1.1}}$ | $\mathbf{56.3{\scriptstyle \pm 0.5}}$ | $\mathbf{60.5{\scriptstyle \pm 2.0}}$ | $\mathbf{59.7{\scriptstyle \pm 2.9}}$ | $\mathbf{42.8{\scriptstyle \pm 0.5}}$ | $\mathbf{37.7{\scriptstyle \pm 0.3}}$ | $\mathbf{36.4{\scriptstyle \pm 0.7}}$ |
>
> We see that ViMON faces challenges in more complex videos (MOVI-{C,D,E}) as the original paper mostly tested it on simple synthetic datasets with only a few objects in a video.
>
> Finally, it is not straightforward to directly compare VONet with MONet as the latter was originally proposed for single-image  scenarios. As suggested by Reviewer 9HJm, we've performed an experiment of comparing VONet with MONet regarding their latency and accuracy on individual video frames of MOVI-{C,D,E}. The conclusion is that  VONet is not only much more efficient in attention mask generation, but also more competent to decompose complex images into objects. Please see our respsonse to Reviewer 9HJm for details.

---

> ### Author Response · Authors · 2023-11-21
> **Response to Reviewer v4jo**
>
> > Can the method in this paper correctly handle objects appearing in the middle of the video or reappearing after being occluded and maintain temporal consistency? The MOVI dataset does not seem to be able to model this situation, consider adding experiments in natural scenes.
>
> MOVI datasets do model these scenarios. We have rendered several demos below to show that VONet is able to generate temporally consistent object masks under occlusion.
>
> In each demo, 'fg_seg' is the foreground objects output by VONet and 'gt_seg' is the groundtruth.
>
> https://i.imgur.com/pAV0bS9.gif
> (The gray cylinder was occluded by the big red cube and appeared as the cube moved away. The small purple ball passed by the small skyblue cube in the back, and was occluded by it and then reappeared.)
>
> https://i.imgur.com/BDxBDAN.gif
> (The gold cylinder was once completely occluded by the red cube in the middle of the video.)
>
> https://i.imgur.com/QCfEQDt.gif
> (The shoe was completely blocked by other objects in the beginning.)
>
> https://i.imgur.com/mSZbOKW.gif
> (The book entered the frame from the top and moved towards the bottom.)
>
> In all these examples, the generated object masks are consistent w.r.t. time (although not in a perfect sense sometimes).
>
> The reason why VONet is able to handle these cases is that, our slot latent, modeled by a GRU, can (in theory) encode information about the history of the slot all the way back to the beginning of the video. When an object appears again in the video frame, the past information contained in the propagated slot latent will make the slot attend to that object.
>
> > Existing video object-centric learning methods based on slot attention are developing rapidly, e.g.DINOSAUR([https://arxiv.org/abs/2209.14860](https://arxiv.org/abs/2209.14860)), VideoSAUR([https://arxiv.org/pdf/2306.04829.pdf](https://arxiv.org/pdf/2306.04829.pdf)), and perform well on natural scene datasets such as YouTube-VIS, please explain the advantages of the method in this paper compared to methods based on slot attention.
>
> The paper already includes DINOSAUR and VideoSAUR as references and we are well aware of their existence. DINOSAUR [Seitzer et al. (2023)] is a SOTA method on object discovery from complex real-world images. Technically, it's a single-image method that cannot handle video data. VideoSAUR [Zadaianchuk et al. (2023)] is an extension to DINOSAUR to handle videos, and it was reviewed in our related work section:
>
> *" ... certain previous works,  such as Zadaianchuk et al. (2023), employ a vision backbone pretrained on large-scale datasets for
> feature extraction from video frames, aiding in the process of object learning. While their objective  remains unsupervised, the use of pretrained features integrates valuable prior knowledge of every-day object appearances.  Consequently, the results they reported, though surpassing what typical unsupervised methods including ours could achieve, hold limited reference value in this context.*"
>
> In a word, pre-trained features obtained from self-supervised methods like DINO [Caron et al. 2021] or MAE [He et al. 2022] are heavily relied on for object discovery by both DINOSAUR and VideoSAUR. These pretrained features representations provide a valuable clue for object discovery from video data. While we totally agree that it is a promising direction to incorporate pretrained prior knowledge into video object learning, the focus of our paper is explore the idea of discovering objects from video in a *tabula rasa* manner, namely, learning from scratch without any pre-existing knowledge or biases, or any kind of supervision such as depth, segmentation or optical flow. This allows us to focus on figuring out what kind of model architecture (instead of prior knowledge) more effectively leads to object emergence from video.
>
> On the other hand, DINOSAUR and VideoSAUR employ standard slot attention techniques [Locatello et al., 2020], with their main innovations lying beyond the enhancement of slot attention itself. Their successes are not mainly attributable to slot attention. Slot attention, produced by a transformer, in general faces a dilemma of attention mask "granularity versus continuity" (fine grained object mask vs. spatially coherent object mask), caused by the attention map resolution. We believe that U-Net can provide a solution to this dilemma thanks to the inductive bias of spatial locality in CNNs. Please see our consolidated response for more details on our motivation of parallel U-Net attention.
>
> Finally, our method VONet, has indeed been empirically compared to several slot-attention methods including SAVI and STEVE, and showed great improvements upon them.
>
> > For writng, the introduction section is short. It is hard to understand the main idea of the whole work.
>
> We've extended the introduction section to elaborate the motivation and contributions of this work. Please see our consolidated response and updated paper.

---

### Author Response · Authors · 2023-11-21
**Consolidated response to all reviewers**

We thank the reviewers' valuable comments and would like to delve deeper into the motivation and contributions of VONet. As appreciated by most reviewers, there are primarily two key innovations of VONet:

1. **Parallel attention based on a U-Net**. This improves the original MONet's recurrent U-Net attention, where the slot attention mask is generated one after another by forwarding the same U-Net multiple times sequentially with updated scopes. This recurrent process makes MONet very inefficient when handling a large number of slots, and impedes its further application to video sequences (see our latency experiment in response to Reviewer 9HJm). Conversely, slot attention [Locatello et al., 2020] is inherently parallelized through a Key-Query-Value attention step for generating slot attention masks. However, our observation is that slot attention usually faces a dilemma which we call "granularity versus continuity" in attention mask generation. Typically, slot attention methods utilize a CNN to extract a feature map of shape $[H, W, C]$ from an input image. This feature map is spatially flattened into a sequence $[HW, C]$ to be queried by each slot to generate an attention mask. Achieving a fine-grained attention mask necessitates choosing larger values for $H$ and $W$. On the other hand, with a large spatial shape, it is difficult to ensure the smoothness of the mask due to the nature of the KQV attention mechanism. The example below demonstrates that with $H=W=64$, STEVE [Singh et al. 2022] (based on slot attention) usually generates jagged slot masks:

https://i.imgur.com/D4xfQVa.png

U-Net can provide a solution to this dilemma thanks to the inductive bias of spatial locality in CNNs. By enabling parallel U-Net attention, VONet can sustain a nearly *constant* attention inference time as the number of slots increases within a reasonable range.

2. **Object-wise sequential VAE framework**. This framework ensures temporal consistency/smoothness of an object between adjacent frames. Moreover, because the predictive prior of the VAE is generated by employing a transformer on the latents of the $K$ slots in the preceding time step (refer to Eq 7), minimizing the KLD between the posterior and the forecasted prior is actually able to model the dynamic interaction and coevolvement of multiple objects within the scene. By adjusting the weight of the KLD, we are able to adjust the importance of temporal consistency relative to video reconstruction quality.

We have revised the paper, with particular emphasis on expanding the introduction section and incorporating additional experiment results. The modifications are indicated by highlighting in orange.

---

### Meta-Review · Area_Chair_Qi7V · 2023-12-06

**Metareview:**

This work proposes a method for video-based object discovery, with two main technical contributions over baselines (namely, the parallel U-net attention module, and the object-wise sequential VAE). There are a similarities between this work and existing ones such as ViMON as pointed out by reviewers, but I find the author response and the additional evaluations satisfactory to demonstrate the performance improvement of this work. This work is evaluated on the 5 subsets of the MOVI dataset (which are synthetically generated, but appear semi-realistic). The proposed method out-performs baselines on MOVI - however, concerns remain on whether this method would generalize to real-world data. Additionally, many concurrent works in this space are starting to show performance on real videos by leveraging pre-trained image features. Despite this, I find the contribution of the method to be sufficient for acceptance but I would encourage the authors to incorporate the feedback from reviewers, and strengthen the contribution through experiments on real-world data if possible.

**Justification For Why Not Higher Score:**

I agree with concerns raised by reviewers on the performance of this method in real-world settings - since all the evaluations are on MOVI, which consists of synthetically generated videos, it is unclear if this method will generalize to more complex settings. Furthermore, leveraging pre-trained visual features such as in DINOSAUR will likely lead to additional performance improvements.

**Justification For Why Not Lower Score:**

This paper evaluates on appropriate baselines and demonstrates convincing performance improvements across all the MOVI dataset subsets. Additionally, many works in this space evaluate on image-based object discovery only, and this work proposes a few technical contributions to achieve better results in the video setting. Furthermore, many more recent works rely on pre-trained features such as DINO for object discovery, and I think it still benefits the field to study how modeling choices might affect how objects may be discovered from video.

---

### Decision · Program_Chairs · 2024-01-16

Accept (poster)